# Standardization of a Sex-Sorting Protocol for Stallion Spermatozoa by Means of Absolute RT-qPCR

**DOI:** 10.3390/ijms241511947

**Published:** 2023-07-26

**Authors:** Erwin Muñoz, Macarena Castro, Luis Aguila, María José Contreras, Fernanda Fuentes, María Elena Arias, Ricardo Felmer

**Affiliations:** 1Laboratory of Reproduction, Centre of Reproductive Biotechnology (CEBIOR-BIOREN), Faculty of Medicine, Universidad de La Frontera, Temuco, P.O. Box 54-D, Chile; e.munoz09@ufromail.cl (E.M.); m.castro09@ufromail.cl (M.C.); luis.aguila@ufrontera.cl (L.A.); mariajose.contrerasr@ufrontera.cl (M.J.C.); f.fuentes10@ufromail.cl (F.F.); mariaelena.arias@ufrontera.cl (M.E.A.); 2Doctoral Program in Applied Cellular and Molecular Biology, Universidad de La Frontera, Temuco P.O. Box 54-D, Chile; 3Master of Science Program with Mention in Biology of Reproduction, Universidad de La Frontera, Temuco P.O. Box 54-D, Chile; 4Department of Agricultural Production, Faculty of Agriculture and Environmental Sciences, Universidad de La Frontera, Temuco P.O. Box 54-D, Chile; 5Department of Agricultural Sciences and Natural Resources, Faculty of Agriculture and Environmental Sciences, Universidad de La Frontera, Temuco P.O. Box 54-D, Chile

**Keywords:** absolute RT-qPCR, PIEZO-ICSI, plasmids, sex-sorting, sperm stallion

## Abstract

Sperm sexing is a technology that can generate great economic benefits in the animal production sector. Techniques such as sex-sorting promise over 90% accuracy in sperm sexing. However, for the correct standardization of the technique, some laboratory methodologies are required. The present manuscript describes in detail a standardized equine sperm sex-sorting protocol using an absolute qPCR-based methodology. Furthermore, the results of absolute qPCR were implemented and validated by generating equine/bovine heterologous embryos by intracytoplasmic sperm injection (ICSI) of presumably sexed equine spermatozoa into bovine oocytes using a piezoelectric system (Piezo-ICSI). Our results indicated that equine sex-sorting spermatozoa had a 97% and 94% certainty for X and Y sperm, respectively, while presumptive female and male equine/bovine hybrid embryos, generated by Piezo-ICSI, had an accuracy of 92% with respect to the desired sex. Therefore, it is concluded that the presented methodology is a reliable, cost-effective, and relatively simple option for standardizing sex-sorting of equine spermatozoa. This is supported by the results of the correct sexing of Piezo-ICSI heterologous embryos generated with the sexed spermatozoa, validating the correct sexing and viability of these gametes.

## 1. Introduction

In the area of animal production, selecting or determining the sex of an animal before it is born can be of great economic benefit [1]. For example, having a thoroughbred foal that is successful in the equestrian arena can be more profitable from a producer’s point of view than a mare with the same characteristics because, although the value of both increases in terms of their progeny, the difference is that a single foal can produce several offspring per year, while a mare can have only one. In mammals, sex is determined by the XY system because males are heterogametic and females are homogametic; this is in contrast to other taxonomic classes, such as birds, where sex is determined by the ZW system, with the female being heterogametic and the male homogametic [2]. Therefore, approaches to sexing offspring in mammals have focused on sexing male gametes, e.g., spermatozoa [3]. Methodologies that have been developed to achieve this goal include immunological detection techniques [4], differences in swimming ability [5], and differential separation by percoll and albumin [6,7], among others. However, the methodology with the best accuracy currently (over 90%) is sexing by flow cytometry or sex-sorting, which is based on the difference in DNA content in Y chromosome with respect to the X chromosome [8]. The data obtained by the horse genome sequencing project show that the X chromosome contains approximately 150 Mb [9], while the Y chromosome could not be correctly sequenced because of its large palindromes and repeated sequences, known as Dark DNA [10,11]. However, this chromosome is estimated to be 50 Mb in size, which is a third less than the female chromosome [10]. Such a difference can be detected by flow cytometry when the DNA of these cells is stained with fluorescent dyes such as DAPI, SYBR Green, or Hoechst.

Parameters such as the voltage used in the cell sorting, the concentration of spermatozoa, and fluorescent dye play a crucial role in achieving a high percentage of accuracy of the technique [8]. In this sense, determining the sex-sorting error is a fundamental step for standardizing these parameters. Therefore, this paper describes a protocol for sex-sorting equine spermatozoa and a methodology based on absolute qPCR to evaluate and standardize flow cytometry parameters during sex-sorting. In addition, to validate the sex-sorting and absolute qPCR results, sexed spermatozoa were used to generate equine/bovine hybrid embryos by piezoelectric intracytoplasmic sperm injection (Piezo-ICSI). Both female and male embryos were 92% accurate with the desired sex, confirming the correct sexing and viability of these spermatozoa.

## 2. Results

### 2.1. Design of Standards Curves

The results of conventional PCR amplification confirmed that pUC19-TNMD and pUC19-SRY vectors were correctly designed as amplicons of 89 and 141 bp, respectively (Figure 1). Dilutions to generate the standards from 3,000,000 to 500,000 copies of each vector showed a directly proportional increase in their Ct of qPCR amplification (Figure 2a,b). The efficiency for both primer pairs was 100%. Therefore, the above results indicate that the standard curves were correctly designed.

### 2.2. Determination of the Sex-Sorting Accuracy

Using absolute RT-qPCR, the number of correctly sexed X and Y cells was quantified, with a SYBR-Green threshold set at a fluorescence value (dR) of 4000. In the case of Y spermatozoa, 2,807,000 of 3,000,000 cells were correctly sexed, which means that 93.57% of the cells carried the Y chromosome, while in the control with unsexed spermatozoa using primers for the Y chromosome, the absolute qPCR indicated that 1,427,000 (47.57%) corresponded to Y spermatozoa. In the case of the X spermatozoa, 96.63% of cells were correctly sexed, while in the control with unsexed spermatozoa, 46.57% corresponded to X spermatozoa. Therefore, both controls with unsexed sperm validated the theoretical calculations, as approximately half of the sperm carried the corresponding chromosome for each pair of primers (Table 1 and Figure 2a,b).

### 2.3. Sexing of Equine/Bovine Hybrid Embryos Produced by PIEZO-ICSI

Equine/bovine hybrid embryos generated by Piezo-ICSI had 82% cleavage assessed at day 3 of culture, with an average development of 8–16 blastomeres (Figure 3). A total of 88 equine/bovine hybrid embryos, 38 embryos injected with presumptive X sperm, and 50 embryos injected with presumptive Y sperm was evaluated by PCR. PCR analysis indicated that 35 (92.1%) of the 38 embryos injected with X sperm were indeed female embryos, while of the 50 embryos injected with Y sperm, 46 (92.0%) were male embryos (Table 1 and Figure 4a). On the other hand, the primers designed for equine sexing did not amplify during PCR in bovine parthenotes, confirming their specificity for *Equus caballus* (Figure 4b).

## 3. Discussion

In the present study, an equine sperm sex-sorting protocol was designed and standardized using a qPCR-based methodology. Although it is based on theoretical calculations that could lead to an error in determining the reliability of sex-sorting, the control with DNA from unsexed spermatozoa showed that this error is negligible. Contrary to Khamlor [12], the simplex qPCR amplification performed here does not seem to show much error compared to multiplex qPCR; thus, the use of probes would not be necessary. Furthermore, the absolute qPCR was validated by the generation of equine/bovine hybrid embryos by microinjection of presumably sexed equine spermatozoa into bovine oocytes by a Piezo-ICSI, with a clear concordance between both results. It is important to highlight that, in this study, the whole embryo was used to determine the sex by PCR; however, in the case of working with equine embryos that are implanted in a recipient mare, there is the alternative of using a biopsy of a single blastomere obtained by micromanipulation to determine the sex of the embryo by PCR, without affecting its viability. This can be achieved using methodologies such as those described by Park et al. [13].

Although there are other techniques to determine the sex-sorting error rate, such as flow cytometric re-analysis [14], multiplex qPCR [12], or PCR on individual spermatozoa [15], the technique reported here is a simpler and cheaper alternative. It is worth noting that this is one of the first papers to focus on determining the error rate of sex-sorting in equine spermatozoa.

Regarding the endemic genes selected to identify each sex chromosome in equine spermatozoa, both were found efficient for this purpose. The SRY gene is frequently used to identify the Y chromosome in different mammalian species [16,17]. However, the TNMD gene had not been previously used for X chromosome identification, since one of the most commonly used genes is PLP [18,19]. Therefore, an alternative to the PLP gene for X chromosome identification is also reported in this study.

As for the designed plasmids used to generate the standards curves for absolute quantification during this work, we linearized the plasmid DNA to prevent overestimates of the results of the absolute qPCR. Although some researchers in scientific forums debate whether it is necessary or not to linearize plasmids for use in absolute qPCR, several studies agree with us, demonstrating that linearization of plasmids using restriction enzymes is necessary to obtain reliable results, as the supercoiling of circular plasmids generate an overestimation of qPCR results [20,21].

During this work, different techniques for DNA extraction from spermatozoa were tested, including commercial kits. Due to the high compaction of sperm DNA by protamines [22], which resulted in a low extraction yield and purity of the genetic material, finding a specific protocol for DNA extraction from spermatozoa was necessary. Universal DNA extraction protocols such as that described by Wang et al. [23] were also tested. However, the methodology with the highest yield and DNA purity was that described by Vázquez [24], confirming that protocols and kits used for DNA extraction from somatic cells may require modification for efficient sperm DNA extraction.

## 4. Materials and Methods

### 4.1. Primers Design

To identify the X and Y chromosomes, regions of genes endemic to each chromosome were amplified by PCR. For the X chromosome, primers were designed to amplify a region of the TNMD gene, which codes for tenomodulin, a protein important for tendon maturation. For the Y chromosome, primers were designed to amplify a region of the SRY gene, which codes for testis determinant factor. Primer design was performed using AmplifX v2.0.7 software, and physicochemical parameters and dimer or hairpin formation were evaluated using IDT’s OligoAnalyzer tool v3.1. Moreover, to eliminate any possibility that the primers amplified a non-specific region within the equine genome, an off-target analysis was performed with Primer-BLAST v2.11.0 software from the NCBI database. The primers designed are shown in Table 2.

### 4.2. Design of Standards for Absolute qPCR

To obtain the standard curves that would be used for absolute quantification, two constructs were designed to simulate both sex chromosomes. First, using the Quick-DNATM Miniprep kit (Zymo Research, Irvine, CA, USA), DNA was extracted from the blood of a foal and a mare, following the manufacturer’s instructions. Then, using AmplifX v2.0.7 software, primers flanking regions of the TNMD and SRY genes were designed and analyzed using OligoAnalyzer v3.1 and Primer-BLAST v2.11.0 software in the same way as explained above. In addition, a restriction site for *KasI* and *BamHI* enzymes was added to the 5’ ends of the primers (see sequences in Table 1). Then, using a high-fidelity DNA polymerase (ThermoFisher, Waltham, MA, USA), the TNMD and SRY gene regions were amplified. The resulting amplicons were digested with *KasI* (NEB, Ipswich, MA, USA) and *BamHI* (NEB, Ipswich, MA, USA) restrictases to leave their ends cohesive. The digested product was visualized on a 1.5% agarose gel electrophoresis, which was run for 1 h at 110 volts. The bands containing the amplicons with their cohesive ends were extracted using the QIAquick^®^ Gel Extraction kit (QIAGEN, Valencia, CA, USA), following the manufacturer’s instructions. 

On the other hand, the pUC19 plasmid was subcloned into commercial DH5α bacteria (ThermoFisher, Waltham, MA, USA), digested by *KasI* and *BamHI* enzymes, and the digestion product was run on agarose gel electrophoresis under the same conditions as above. Finally, the TNMD and SRY gene fragments were separately ligated into the cohesive pUC19 vector, resulting in the pUC19-TNMD and pUC19-SRY plasmids, respectively (Figure 5).

We confirmed by conventional PCR that the vectors were correctly constructed, using the primers designed for qPCR. The resulting amplicons were visualized on 2.5% agarose gel electrophoresis run at 110 volts for 1 h (Figure 4).

### 4.3. Collection of Semen Samples and Cryopreservation

Semen was collected from three Chilote stallions aged 3–8 years with a Colorado artificial vagina during the reproduction season in compliance with the Scientific Ethics Committee of the Universidad de La Frontera (Act N° 057/2016). Immediately after collection, the semen was diluted by adding commercial diluent (Botusemen^®^, Botucatu, Brazil) in a 1:1 ratio and transported to the laboratory at 37 °C within 15 min of collection. Sperm concentration was assessed using a Neubauer chamber by diluting the sample 100-fold. The diluted ejaculates were then centrifuged at 1000 RPM for 10 min, the supernatant was removed, and the cell pellet was brought to a concentration of 200 million/mL using Equiplus Freeze 1 step^®^ commercial cryopreservation medium (Minitube, Valencia, Spain). Afterwards, the samples were stored in 0.5 mL straws (Minitube, Valencia, Spain), heat-sealed, and balanced at 6 °C for 1 h. Finally, the straws were exposed to liquid nitrogen (LN2) vapors for 10 min, plunged, and stored in LN2 at −196 °C for later use.

### 4.4. Sex-Sorting of Equine Sperm

Sperm for sex-sorting were thawed for 30 s in a thermoregulated bath at 37.5 °C, followed by two washes with calcium- and magnesium-free PBS (ThermoFisher, Waltham, MA, USA), supplemented with 10% bovine serum albumin (BSA) and centrifuged at 1200 RPM for 5 min, to remove the cryopreservation medium. The spermatozoa were brought to a concentration of 15 × 10^6^ SPZs/mL and stained with Hoechst 34,580 (ThermoFisher, Waltham, MA, USA) at a final concentration of 800 ng/mL, incubated for 20 min in the dark. Sex-sorting was performed on a Cell Sorter FACS ARIA FUSION cytometer (Becton Dickinson, San Jose, CA, USA), equipped with a 405 nm laser, which was used to irradiate Hoechst 34580. The presumptive X and Y sperm populations were chosen according to their fluorescence intensity (Figure 6). Sperm were analyzed at a rate of 3000–4000 cells per sec, and the sorted sperm (predominantly X-bearing or predominantly Y-bearing) were collected at a rate of approximately 34–36 cells per sec. For sperm separation, 397 volts and a 70 µm/psi injector were used. Each sperm population was collected separately in cytometry tubes containing 500 µL of PBS to minimize damage from the sperm’s impact on the tube. Data were acquired and analyzed using FACSDiva™ v8.0.2 software (Becton, Dickinson and Company).

At the end of sex-sorting, spermatozoa were suspended at a concentration of 90,000 SPZs/mL in calcium and magnesium-free PBS. Then, the PBS containing spermatozoa was supplemented with 10% molecular biology grade BSA (ThermoFisher, Waltham, MA, USA) and centrifuged for 10 min at 1200 RPM. Finally, using a 200 μL pipette and taking great care not to disturb the sometimes inconspicuous sperm pellet, the supernatant was discarded to concentrate sexed spermatozoa to a final volume of 50 μL.

### 4.5. DNA Extraction from Equine Spermatozoa

Due to the high compaction of DNA in sperm cells by proteins called protamines [22], DNA was extracted using the methodology described by Vázquez [24], with some modifications. The membrane was removed from the spermatozoa by incubating them for 10 min at 30 °C in 10% CTAB (cetyltrimethylammonium bromide). Cells were then lysed using a solution containing Proteinase K 1 mg/mL, 1% SDS, 1% DTT, and 0.025% DMSO in 50 mM Tris HCl (pH 8.9), incubated for 30 min at 55 °C. Two washes were performed with 50 mM Tris HCl (pH 8.9) at 11,000 RPM for 5 min. The DNA was then decondensed using a solution containing DTT (180 mM final) and Heparin (1 U/mL final). Again, two washes were performed with 50 mM Tris HCl (pH 8.9) at 11,000 RPM for 5 min, and the resulting pellet was resuspended in 500 μL of 50 mM Tris HCl (pH 8.9). Then, 300 μL of ice-cold acetone was added, followed by centrifugation at 5000 RPM for 5 min, discarding the supernatant. The pellet was resuspended in 300 μL of absolute ethanol and centrifuged at 5000 RPM for 5 min to remove the supernatant. Finally, the DNA was reconstituted in molecular biology grade water. Nucleic acid quantification was performed using the Qubit 3.0 fluorometric quantifier (ThermoFisher, Waltham, MA, USA), following the manufacturer’s instructions.

### 4.6. Determination of Sex-Sorting Error Rate

To determine the percentage error of the sex-sorting, the number of copies of the pUC19-TNMD and pUC19-SRY vectors in certain pg of DNA was first calculated using the dsDNA copy number calculator v1.29.04 software, using the formula “number of copies = (amount of DNA in ng × 6.022 × 10^23^)/(size in bp × 1 × 10^9^ × 650)”; in this way, the standard curves were designed for the quantification of both sex chromosomes (Table 3).

On the other hand, to know the number of spermatozoa to which a given amount of μg of DNA corresponds, the same formula was used, calculating that an X spermatozoon has 2.46 Gb (2,460,000,000 bases) in its genome and a Y spermatozoon has 2.35 Gb (2,350,000,000 bases). This calculation was performed using data from the horse genome sequencing project found in the NCBI database under ID: 145 [9]. Finally, for absolute qPCR quantification, 7.96 and 7.6 μg of DNA from X and Y (presumably sexed) spermatozoa, respectively, which would be equivalent to 3,000,000 cells, were added to determine the sex-sorting error, noting that each spermatozoon, being a haploid cell, will contain only one copy of the TNMD or SRY gene. In addition, the same amount of DNA was added, but from unsexed spermatozoa, to validate the theoretical calculations.

As the table indicates, the standards were performed using the vectors designed according to the number of copies or molecules of each plasmid to compare it with the number of presumed sexed spermatozoa. The calculation of how many molecules are in certain picograms of plasmid was performed with the software dsDNA copy number calculator v1.29.04, which considers Avogadro’s number in the following formula: Number of molecules = (amount (in pg) × 6.022 × 10^23^)/(size (in bp) × 1 × 10^9^ × 650).

### 4.7. Absolute RT-qPCR

Absolute qPCR quantification was performed on an Agilent Technologies Stratagene Mx3000P thermal cycler using Brilliant II SYBR Green mastermix (Agilent Technologies, Santa Clara, CA, USA). The reaction conditions were: 10 min at 95 °C, 30 cycles of 20 s at 95 °C, 20 s at 55 °C, and 20 s at 72 °C, ending with an incubation of 1 s at 25 °C and 15 s at 72 °C. The six standards for each chromosome (pUC19-TNMD and pUC19-SRY) linearized with the *KasI* enzyme, DNA extracted from presumably sexed spermatozoa, and two controls with unsexed spermatozoa (one for each pair of primers) were amplified. All were performed in triplicate. The data were analyzed by MxPro v4.10 software (Agilent Technologies).

### 4.8. Oocyte Collection and In Vitro Maturation (IVM)

To validate the results of absolute qPCR of the sperm sexing, a heterologous ICSI assay was performed by microinjection of equine sperm into bovine oocytes. Importantly, the designed primers do not have homologous sites across the bovine genome; therefore, the heterologous embryos will have only one alignment site for the designed primers.

Bovine ovaries were collected from a local slaughterhouse (Frigorifico Temuco, Temuco, Chile). Cumulus–oocyte complexes (COCs) were aspirated from 2–7 mm follicles using a 12-gauge disposable needle. COCs with several cumulus cell layers and homogeneous cytoplasm were matured in TCM-199 medium (Gibco™, Waltham, MA, USA) supplemented with 10% (*v*/*v*) inactivated bovine fetal serum (BFS; Gibco™, Waltham, MA, USA), LH hormone (6 μg/mL), and estradiol (1 μg/mL). In vitro oocyte maturation was performed for 22–24 h at 38.5 °C in a humidified atmosphere at 5% CO_2_ [25]. Matured oocytes were stripped of cumulus cells by vortexing for 5 min in the presence of hyaluronidase 1 mg/mL and selected based on the presence of the first polar body.

### 4.9. Heterologous ICSI, Bovine Parthenotes Production, and Embryo Culture

ICSI was carried out as previously described [26] using Narishige manipulators (Narishige Scientific Instrument Lab) mounted on an inverted microscope with Hoffman optics (Eclipse TS100F; Nikon Instruments, New York, NY, USA). Piezo-ICSI was performed in Hamster Embryo Culture Medium–Hepes buffered (HECM–HEPES) at 38.0 °C. One part of the sperm suspension was mixed with one part injection buffer containing 5% polyvinylpyrrolidone (PVP, Sigma-Aldrich, St. Louis, MO, USA). Immediately prior to injection, motile spermatozoa were immobilized by applying a few piezo pulses to the sperm tail. Equine sperm were delivered into the bovine ooplasm using a piezo micropipette-driving unit (PMM-4g Piezo drill, Prime Tech, Ibaraki, Japan).

On the other hand, to evaluate by PCR that the primers designed for equine sexing do not present specific amplification sites in the *Bos taurus* genome, bovine parthenogenetic embryos were produced as previously described [25]. Briefly, mature bovine oocytes were activated by 5 min exposure to 5 µM ionomycin (Calbiochem, San Diego, CA, USA) followed by 5 h of incubation in KSOM embryo culture medium containing 1 μg/mL of anisomycin [25].

For in vitro culture, groups of 25 presumptive embryos were placed in droplets (50 µL) of KSOM embryo culture medium under mineral oil. The embryo culture dishes were incubated in a gas mixture containing 5% CO_2_, 5% O_2_, and 90% N_2_ at 38.5 °C in saturation humidity for 3 days.

## 5. Conclusions

The methodologies presented here can be used to sex equine spermatozoa, evaluate the percentage error of sex-sorting, and standardize the parameters of flow cytometry for each particular case. This was confirmed in the present study because the percentage of embryos of the correct sex, with the sex chromosome of the microinjected spermatozoa for oocyte fertilization, is concordant with the results of absolute qPCR. However, for commercial reasons or fertility characteristics of each stallion, it may be necessary to modify some parameters of the flow cytometer, such as the concentration of the spermatozoa or the concentration of the fluorescent stain. This will require a re-standardization of the sex-sorting, for which the absolute qPCR methodology presented here is a fast, accurate, and economical alternative, since it does not require antibodies or fluorescent probes, and is relatively simple once it has been implemented in the laboratory.

## Figures and Tables

**Figure 1 ijms-24-11947-f001:**
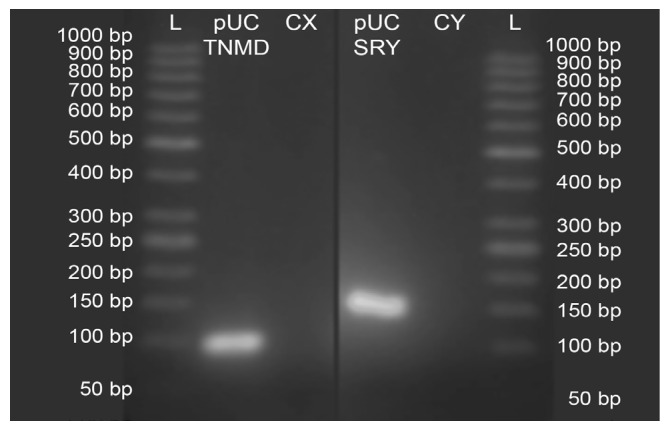
Confirmation of the correct design of the pUC19-TNMD and pUC19-SRY vectors by PCR. Lanes L correspond to the DNA Ladder “GeneRuler 50 bp DNA Ladder (ThermoFisher, Waltham, MA, USA)”. Lane pUC-TNMD corresponds to the pUC19-TNMD vector with the qPCR-TNMD primers. Lane pUC-SRY corresponds to the vector pUC19-SRY with the qPCR-SRY primers. The CX lane corresponds to the pUC19-TNMD vector with the qPCR-SRY primers and the CY lane corresponds to the pUC19-SRY vector with the qPCR-TNMD primers.

**Figure 2 ijms-24-11947-f002:**
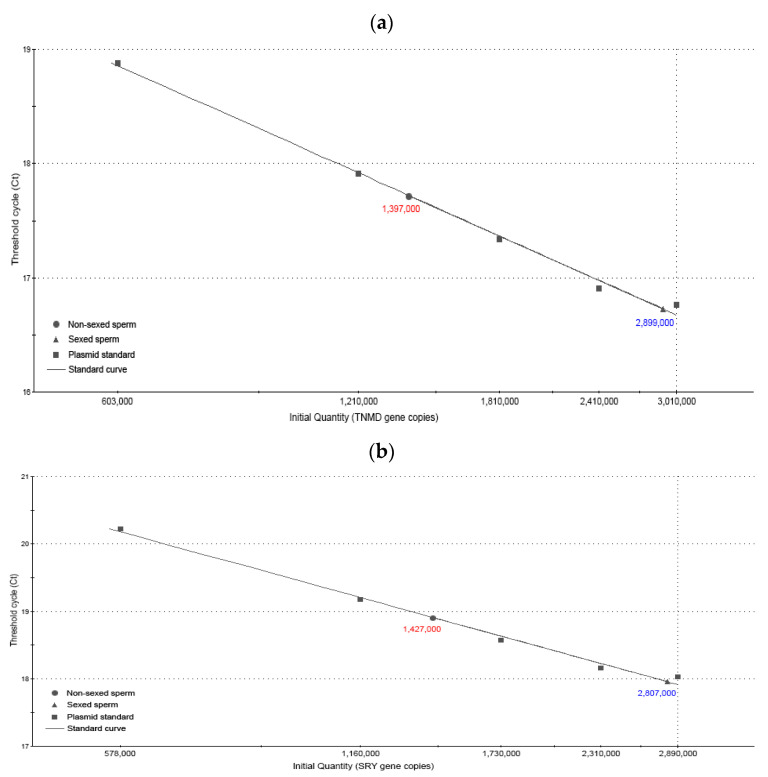
Standard curves to determine the percentage error of sex-sorting. (**a**) The curve of standards made with the pUC19-TNMD plasmid and the number of spermatozoa with a copy of the TNMD gene that carries the X chromosome are shown. (**b**) The curve of standards made with the pUC19-SRY plasmid and the number of spermatozoa with a copy of the SRY gene that carries the Y chromosome are shown. Squares represent the standards made with the plasmids, the triangles represent the sexed spermatozoa, and the circles represent the unsexed spermatozoa, in both groups. In red is the control of unsexed sperm and blue is sexed sperm samples in both graphs.

**Figure 3 ijms-24-11947-f003:**
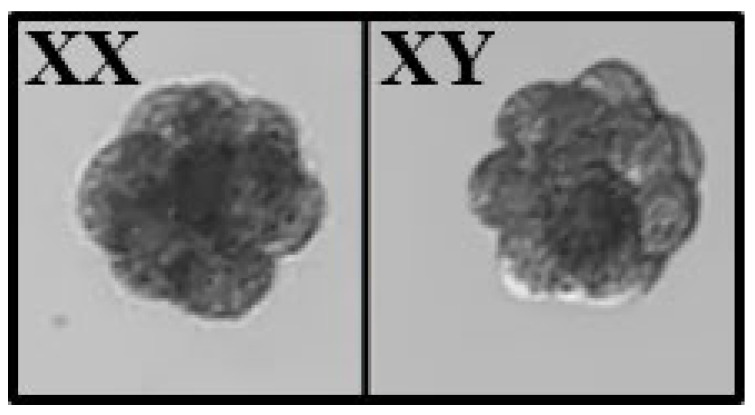
Equine/bovine hybrid embryos produced by Piezo-ICSI. Representative images of equine/bovine hybrid embryos on day 3 of culture. Both embryos are in the morula stage with 8–16 blastomeres.

**Figure 4 ijms-24-11947-f004:**
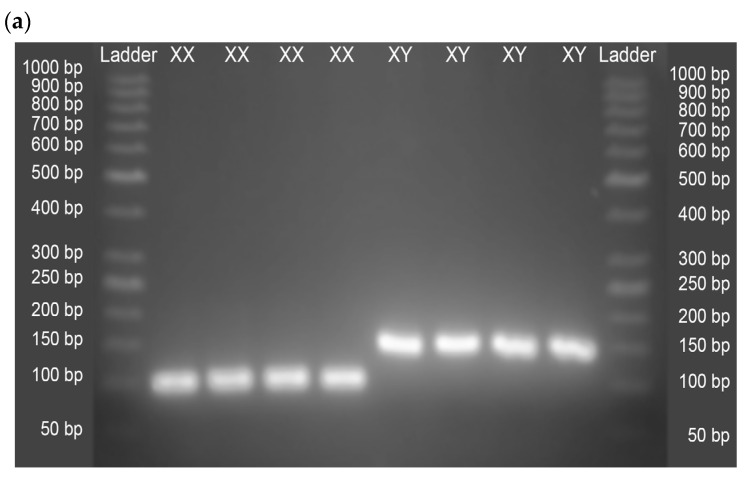
PCR sexing of equine/bovine hybrid embryos generated by Piezo-ICSI. Lanes L correspond to the DNA Ladder GeneRuler 50 bp DNA Ladder (ThermoFisher, Waltham, MA, USA). Lanes XX and XY correspond to the sexing of female and male embryos, respectively. Lanes CX and CY correspond to the confirmation that the qPCR-TNMD and qPCR-SRY primers have no amplification sites in the bovine parthenotes genome. (**a**) Representative electrophoresis of the PCR sexing of 4 female embryos (89 bp) and 4 male embryos (141 bp), produced with the respective sexed spermatozoa, can be observed. (**b**) Representative electrophoresis shows that primers qPCR-TNMD and qPCR-SRY do not generate an amplicon by PCR, with the DNA of bovine parthenotes as template.

**Figure 5 ijms-24-11947-f005:**
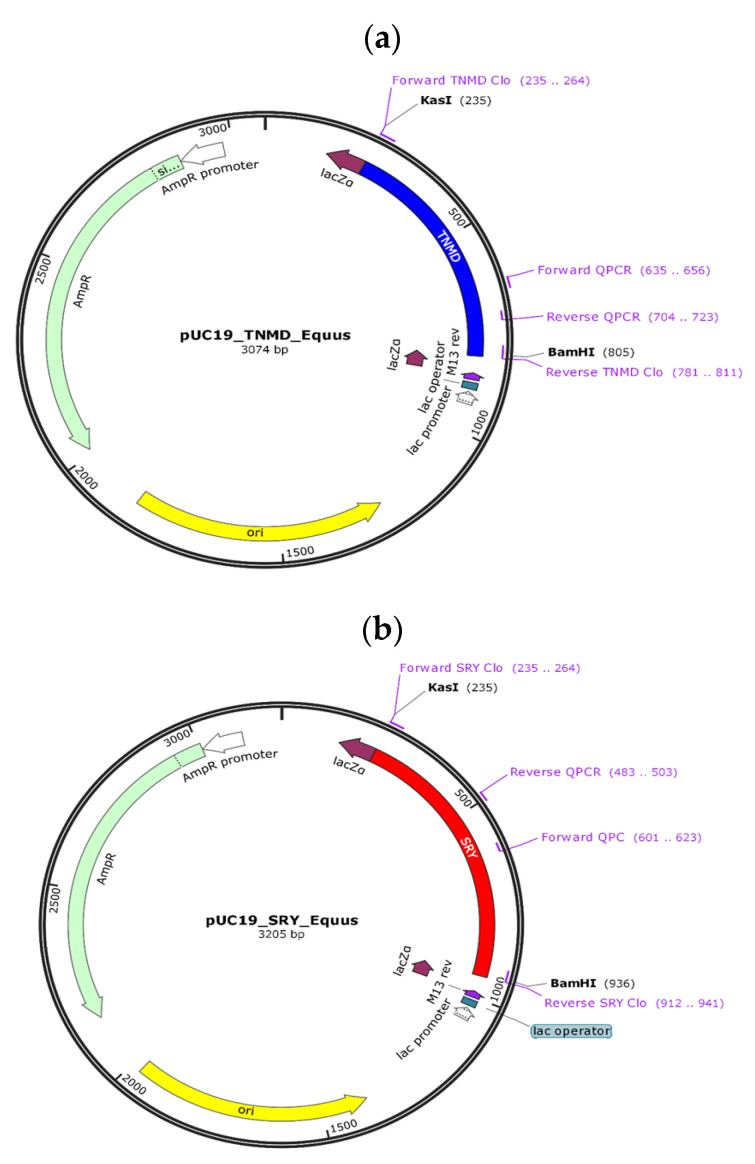
pUC19-TNMD and pUC19-SRY vectors. The two constructs designed for making the standards for absolute qPCR quantification are shown. (**a**) pUC19-TNMD vector contains a fragment of the TNMD gene, endogenous to the X chromosome. (**b**) pUC19-SRY vector containing a fragment of the SRY gene, endogenous to the Y chromosome.

**Figure 6 ijms-24-11947-f006:**
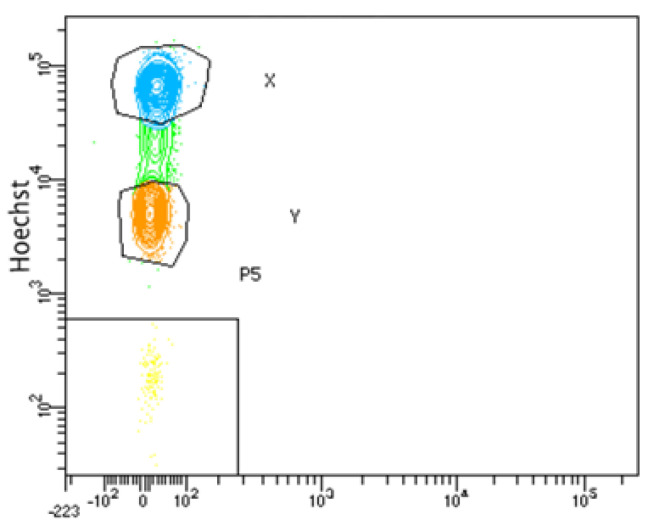
Dot Plot of sperm populations. Representative image showing the two sperm populations identified by the differences in fluorescence of the X and Y chromosomes during sex-sorting by flow cytometry. In blue is the X sperm population, orange is the Y sperm population, green is indetermined sperm population and yellow is debris.

**Table 1 ijms-24-11947-t001:** Percentage of correct sexing of sperm and embryos.

Sexing	Primers Used	% of Correct Sex
Absolute RT-qPCR of sexed spermatozoa X	qPCR-TNMD	96.63%
Absolute RT-qPCR of unsexed spermatozoa	qPCR-TNMD	46.57%
Absolute RT-qPCR of sexed spermatozoa Y	qPCR-SRY	93.57%
Absolute RT-qPCR of unsexed spermatozoa	qPCR-SRY	47.57%
PCR of equine/bovine embryos generated with a sperm X	qPCR-TNMD	92.1%
PCR of equine/bovine embryos generated with a sperm Y	qPCR-SRY	92.0%

Percentage of correct sex of sexed spermatozoa and embryos produced by PIEZO-ICSI using sexed spermatozoa.

**Table 2 ijms-24-11947-t002:** Sequences of designed primers.

Name	Sequence 5′ to 3′	Use	Amplicon
qPCR-SRY-F	AGGACAGCAACATACCGTTCTCG	absolute qPCR of sperm embryo sexing.	141 bp
qPCR-SRY-R	GCATTCATGGGTCGTTTGACA	absolute qPCR of sperm embryo sexing.	141 bp
qPCR-TNMD-F	GCGGGTTATCTGTCGTGTCATC	absolute qPCR of sperm, embryo sexing.	89 bp
qPCR-TNMD-R	GCTCTTGTGCTCGAACTTGC	absolute qPCR of sperm, embryo sexing.	89 bp
SRY-F	GCAATGGCGCCCGGGAAGCGGTTTGTCACTTTTCT	Cloning of the SRY gene from *Equus caballus*.	695 bp
SRY-R	AACTAGGATCCTGGGGATTAGAAGTAGGGCACAGA	Cloning of the SRY gene from *Equus caballus*.	695 bp
TNMD-F	GCAATGGCGCCTCACAGCCCCTCAGATCAAAGTCA	Cloning of the TNMD gene from *Equus caballus*	564 bp
TNMD-R	AACTAGGATCCGCTACCAGGAGCCAAATGCCTTAT	Cloning of the TNMD gene from *Equus caballus*.	564 bp

The grey color indicates the restriction site of the *KasI* restrictase. The olive color indicates the restriction site of the *BamHI* restrictase.

**Table 3 ijms-24-11947-t003:** Standards curves.

Quantity of Standard pUC19-SRY	Number of Molecules of pUC19-SRY Vector	Quantity of Standard pUC19-TNMD	Number of Molecules of pUC19-TNMD Vector
10 pg	2,890,000	10 pg	3,010,000
8 pg	2,310,000	8 pg	2,410,000
6 pg	1,730,000	6 pg	1,810,000
4 pg	1,160,000	4 pg	1,210,000
2 pg	578,000	2 pg	603,000
0 pg	0	0 pg	0

## Data Availability

The data presented in this study are available on request from the corresponding author.

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
