# Peer review of "Standardization of a Sex-Sorting Protocol for Stallion Spermatozoa by Means of Absolute RT-qPCR"

_ijms, 2023, doi:10.3390/ijms241511947_

Round 1

Reviewer 1 Report

Title: Standardization of a Sex-Sorting Protocol for Stallion Sperma-tozoa by means of Absolute RT-qPCR

Journal: Int. J. Mol. Sci.

Manuscript ID: ijms-2465149

The manuscript highlights on a standardized equine sperm sex-sorting protocol using absolute qPCR-based methodology. These results are critical in the field of reproductive biotechnology and might be beneficial in developing the methodology of the sex-sorting and absolute qPCR result. I invite the authors to read their articles carefully and improved it. I have recommended Major revision.

My comments are the following:

-        Authors used the old version of the template for their manuscript, please change with updated one.

-        In all the paper authors should use sperm or spermatozoa, its not correct to use the words sperms for plural.

-        The manuscript should be lined

-        All figures should be re-established again.

-        Some spaces between the words within the same line were observed. (Fig 2a,b),

-        Affiliation 2; e-mail@e-mail.com deleted.

-        This sentence needs rewritten, techniques such as sex-sorting promise up to 90% accuracy when sexing spermatozoa.

-        Reduced the words “The results of the absolute” in the abstract.

-        Deleted embryo tested

-        4.8. Arias et al., 2016 corrected please

-        Figure 5. Should be adjusted, some parts aren’t appeared.

-        2.2. Determination of the sex-sorting error rate should be changed to “Determination of the sex-sorting accuracy “.

-        those described by Park [17] should be Park et al. [17]

-        All citations must be revised.

-        All references are elder.

-        The title of table 1 is not clear.

-        Introduction needs some data regarding the embryos results after sexing sperm.

-        In vitro or in vivo should be italic in all the manuscript.

-         

-         

moderate English Editing is required 

Author Response

Answer to reviewer 1

The manuscript highlights on a standardized equine sperm sex-sorting protocol using absolute qPCR-based methodology. These results are critical in the field of reproductive biotechnology and might be beneficial in developing the methodology of the sex-sorting and absolute qPCR result. I invite the authors to read their articles carefully and improved it. I have recommended Major revision.

Thank you very much for your comments, we have reviewed our article and considered your corrections as detailed below:

-Authors used the old version of the template for their manuscript, please change with updated one.

Response: We have updated the template now.

-In all the paper authors should use sperm or spermatozoa, its not correct to use the words sperms for plural.

Response:  Corrected

-The manuscript should be lined.

Response: Corrected.

-All figures should be re-established again.

Response: Corrected.

-Some spaces between the words within the same line were observed. (Fig 2a,b),

Response: Corrected.

-Affiliation 2; e-mail@e-mail.com deleted.

Response: Corrected.

-This sentence needs rewritten, techniques such as sex-sorting promise up to 90% accuracy when sexing spermatozoa.

Response: The sentence has been corrected “Techniques such as sex-sorting promise over 90% accuracy in sperm sexing.”

-Reduced the words “The results of the absolute” in the abstract.

Response: Corrected

-Delete embryo tested

Response: Corrected

-4.8. Arias et al., 2016 corrected please

Response: Citation in the manuscript has been corrected.

-Figure 5. Should be adjusted, some parts aren’t appeared.

Response: Figure 5 has been adjusted.

-2.2. Determination of the sex-sorting error rate should be changed to “Determination of the sex-sorting accuracy “.

Response: Corrected Title 2.2.  as suggested.

-those described by Park [17] should be Park et al. [17]

Response: Citation in the manuscript has been corrected.

-All citations must be revised.

Response: Corrected. All citations have been revised.

-All references are elder.

Response: Corrected. Some references have been updated.

-The title of table 1 is not clear.

Response: Title of table 1 has been corrected.

-Introduction needs some data regarding the embryos results after sexing sperm.

Response: Corrected. Data on the embryo results after sexing sperm have been added in the introduction.

-In vitro or in vivo should be italic in all the manuscript.

Response: Corrected.

-Moderate English Editing is required

Response: Thank you for your suggestion. English has been improved.

Reviewer 2 Report

The purpose of this study was to depicts a standardized equine sperm sex-sorting protocol using absolute qPCR-based methodology. Besides, the results of absolute qPCR were implemented and validated by generating equine/bovine heterologous embryos by intracytoplasmic sperm injection (ICSI) of presumably sexed equine spermatozoa into bovine oocytes using a piezoelectric system (Piezo-ICSI).

The results concluded that the presented absolute qPCR-based methodology is a reliable, cost-effective, and relatively simple option to standardize sex-sorting of equine spermatozoa.

While the results and conclusions are not novel, they are based on solid data and help to extend the knowledge on the topic, which may be beneficial in the future.

Therefore, I have no further comments against the manuscript.

Author Response

Answer to reviewer 2

There are no requirements raised by reviewer 2

Round 2

Reviewer 1 Report

I have accepted this manuscript.

1- the following sentence did not changed, please change it  

Int. J. Mol. Sci. 2021, 22, x FOR PEER REVIEW.

2- please, citations should used the abbreviations of journals.